# Depression, depressive symptoms and treatments in women who have recently given birth: UK cohort study

Irene Petersen,[1,2] Tomi Peltola,[1,3] Samuel Kaski,[3] Kate R Walters,[1] Sarah Hardoon[1]

[1]Department of Primary Care and Population Health, University College London, London, UK
[2]Department of Clinical Epidemiology, Aarhus University, Aarhus, Denmark
[3]Department of Computer Science, Helsinki Institute for Information Technology HIIT, Aalto University, Espoo, Finland

**Correspondence to**
Professor Irene Petersen;
i.petersen@ucl.ac.uk

## ABSTRACT

**Objectives** To investigate how depression is recognised in the year after child birth and treatment given in clinical practice.

**Design** Cohort study based on UK primary care electronic health records.

**Setting** Primary care.

**Participants** Women who have given live birth between 2000 and 2013.

**Outcomes** Prevalence of postnatal depression, depression diagnoses, depressive symptoms, antidepressant and non-pharmacological treatment within a year after birth.

**Results** Of 206 517 women, 23 623 (11%) had a record of depressive diagnosis or symptoms in the year after delivery and more than one in eight women received antidepressant treatment. Recording and treatment peaked 6–8 weeks after delivery. Initiation of selective serotonin reuptake inhibitors (SSRI) treatment has become earlier in the more recent years. Thus, the initiation rate of SSRI treatment per 100 pregnancies (95% CI) at 8 weeks were 2.6 (2.5 to 2.8) in 2000–2004, increasing to 3.0 (2.9 to 3.1) in 2005–2009 and 3.8 (3.6 to 3.9) in 2010–2013. The overall rate of initiation of SSRI within the year after delivery, however, has not changed noticeably. A third of the women had at least one record suggestive of depression at any time prior to delivery and of these one in four received SSRI treatment in the year after delivery. Younger women were most likely to have records of depression and depressive symptoms. (Relative risk for postnatal depression: age 15–19: 1.92 (1.76 to 2.10), age 20–24: 1.49 (1.39 to 1.59) versus age 30–34). The risk of depression, postnatal depression and depressive symptoms increased with increasing social deprivation.

**Conclusions** More than 1 in 10 women had electronic health records indicating depression diagnoses or depressive symptoms within a year after delivery and more than one in eight women received antidepressant treatment in this period. Women aged below 30 and from the most deprived areas were at highest risk of depression and most likely to receive antidepressant treatment.

## INTRODUCTION

Many women experience depression in the year after they have given birth. Postnatal depression affects an estimated 10%–19% of women, although the estimates vary substantially between countries and settings.[1–4] Depression may have severe consequences for

## Strengths and limitations of this study

► A major strength of this study is that we have access to a very large sample of primary care electronic health records of women who gave live birth.
► These records reflect clinical practice in UK primary care and are made prospectively.
► We considered a broad definition of depression on clinical evaluation in the year after delivery as there were no specific guidelines to how it should be recorded.
► This study may overestimate the number of women with postnatal depression compared with estimates based on a diagnostic interview and specific diagnostic instruments.
► Non-pharmacological treatment may not be well recorded in primary care electronic health records.

the mother and, in turn, have physical, cognitive and emotional effects on their children's development, continuing into later life.[5–8] A report published by the London School of Economics estimated that perinatal depression, anxiety and psychosis carry a total long-term cost to society of about £6.6 billion for each 1-year cohort of births in the UK.[5] This is equivalent to a cost of just under £10 000 for every single birth in the country. Nearly three-quarters (72%) of this cost relates to adverse impacts on the child rather than the mother.[5]

Guidelines in both the USA and UK on antenatal and postnatal mental health recommend that healthcare professionals should consider asking simple screening questions about current and past histories of depression, anxiety, alcohol and illicit drug use as part of a general discussion about mental health and well-being in pregnancy and the perinatal period.[9 10] However, very limited information is available on when depression is recognised and how it is treated in clinical practice in the year after women have given birth. For most women who experience depression in this period, primary care physicians would be a first point of contact.

In this study, we sought to obtain an overview of actual clinical practice in UK primary care by examining electronic health records on more than 200 000 women who have given live birth between 2000 and 2013. We followed the women for a year after delivery, and our aim was to examine how and when depression and depressive symptoms were recorded and treatment provided in general practice and the inter-relation between antidepressant and non-pharmacological treatment.

## METHODS
### Data source
We used data from The Health Improvement Network. This is a large primary care database that provides anonymised longitudinal general practice (family practice) data on patients' clinical and prescribing records and includes data from around 6% of the UK population. Diagnoses and symptoms are recorded by practice staff using Read codes which is a hierarchical coding system including more than 100 000 codes.[11 12] The Read code system can be mapped to International Classification of Diseases, tenth revision (ICD-10), but in addition the Read codes include a number of symptoms and administrative codes.[12] Prescriptions are issued electronically and directly recorded on the general practice computer systems. In addition, the database holds individual patient-level information about year of birth, date of registration, date of death and transfer out of the practice and information about social deprivation (quintiles of Townsend Deprivation Index scores). The Townsend scores are based on census data (2011) for car ownership, owner-occupation, overcrowding and unemployment in a patient's postcode.[13]

Over 98% of the UK population are registered with a general practitioner (GP),[14] and the UK primary care databases are broadly representative of the UK population.[15 16] While perinatal care is often shared between general practice staff and midwives, the GP remains responsible for women's general medical care including continued prescribing of medicines such as antidepressants. Some women may also receive care from local National Health Service mental health trusts, but trusts have limited prescribing budgets and for most women prescribing of psychotropic medication remains with the GP. Furthermore, after a few weeks after delivery, the care by the midwife ends and GPs are the first point of contact. Typically, women will consult their GP for a postnatal maternal check-up at 6–8 weeks after delivery.

### Study population
We used data from women who have given live birth between 1 January 2000 and 31 December 2013 and who were permanently registered with the same general practice for at least 1 year after delivery. As some women had more than one pregnancy and the risk of postnatal depression may be strongly correlated within women, we randomly selected one pregnancy per woman for our analyses.

### Variables
We identified women with one or more records entered as a Read code in their primary care electronic health records which suggested they had depression, postnatal depression or symptoms of depression as well women on antidepressant and non-pharmacological treatment (referral to counselling and psychotherapy) in the year after they have given birth. Antidepressant treatment was classified as selective serotonin reuptake inhibitors (SSRI), tricyclic antidepressants (TCA) and other antidepressants. For TCA, we only considered treatment that was prescribed above treatment threshold for depression (eg, for amitriptyline we considered prescriptions of 75 mg and above), as lower doses may be prescribed for other reasons such as chronic pain. In addition, we included information on calendar year of delivery, age at delivery and social deprivation.

### Data analysis
First, we estimated the prevalence of any records directly suggestive of depression (postnatal depression, depression diagnoses, depressive symptoms) as well as separate estimates for postnatal depression, depression diagnoses, depressive symptoms, antidepressant or non-pharmacological treatments within a year after giving birth. These estimates are reported in figure 1A. We then estimated how the records were inter-related. Inter-relations were reported as conditional frequencies, that is, the frequency of having a record of X given that one has a record of Y. These frequencies are reported in figure 1B. For example, the figure illustrates that 82% of those who had a diagnosis of depression also had a prescription of an SSRI. On the other hand, 31% of those who had a prescription of SSRI had a diagnosis of depression.

We estimated the timing of the recording within the follow-up year and report cumulative incidence curves (as one minus the Kaplan-Meier estimate). We also estimated smoothed daily hazards using a Gaussian process model[17] to visualise the daily changes in the timing of recording.

For each of the three depression outcomes (depression diagnosis, postnatal depression diagnosis and depression symptoms) and for SSRI and non-pharmacological treatments, we used Poisson regression to model relative risks of having a record associated with age, calendar time and social deprivation (Townsend scores). Age was split into six age groups (15-19, 20-24, 25-29, 30-34, 35-39, 40-49) and calendar time into three periods (2000–2004, 2005–2009, 2010–2013). 95% CI were computed using modified Poisson regression accounting for the clustering of women in general practices. We conducted supportive analyses stratified: (1) on whether women had any record suggestive of depression or treatment prior to delivery, (2) on whether the women had early or late records of depression or treatment. In the latter analyses, we categorised women into two groups: women who had a record

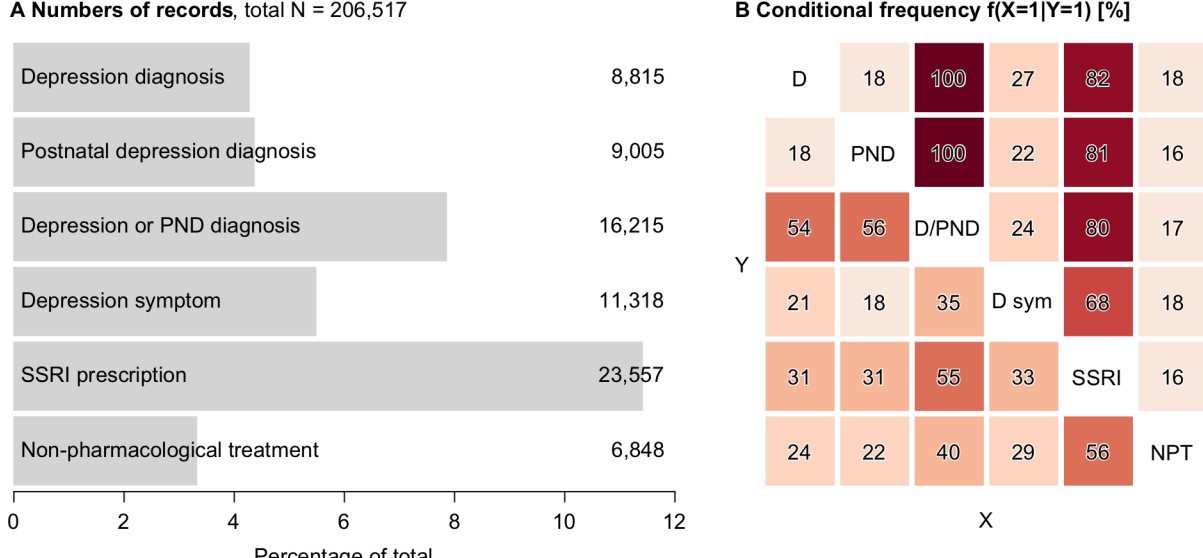

**Figure 1** (A) Numbers of records of depressive diagnoses and symptoms, as well as treatment. (B) Conditional frequency of records: given that one has the condition on the y-axis, what is the frequency of having the condition on x-axis. For example, the figure illustrates that 82% of those who had a diagnosis of depression also had a prescription of an SSRI. On the other hand, 31% of those who had a prescription of SSRI had a diagnosis of depression. D/PND, either or both. D, depression diagnosis; D sym, depression symptom; NPT, non-pharmacological; PND, postnatal depression diagnosis; SSRI, selective serotonin reuptake inhibitor prescription.

of depression or treatments before 42 days after delivery were considered as having an *early* record and women who had a record of depression or treatments after 42 days of delivery were considered as having a *late* record. We investigated whether this was associated with age, social deprivation, calendar time and any record suggestive of depression or treatment prior to delivery using logistic regression.

### Patient and public involvement

Charlotte Walker, who is a mental health service user, has been involved with the original design of the study proposal and provided feedback on this manuscript and thus helped to shape the discussion of the paper from a service user's perspective.

### RESULTS

In total, 206 517 women were included in the study, and there were 23 623 (11%) with at least one record directly suggestive of depression (depression, postnatal depression or symptoms of depression) in the year after delivery. Of these women, there were 4% with a record of depression, 4% with a record of postnatal depression and 5% with symptoms of depression (figure 1A). Of those women with a depression diagnosis, 2349/8815 (27%) also had depressive symptoms (figure 1B), and of those with postnatal depression diagnosis, 2005/9005 (22%) also had depressive symptoms (figure 1B). In contrast, there were 7408/11 318 (65%) women with a record of depressive symptoms *without* either a depression diagnosis or postnatal depression diagnosis.

The number of women with a record suggestive of depression continued to rise throughout the first year after delivery (figure 2). However, the recording of postnatal depression levelled off after the first 3–4 months (figure 2A). For all types of records, there were some clear peaks in recording immediately after delivery and in the period between 6 and 8 weeks after delivery coinciding with the time of postnatal maternal check-up consultation (figure 2A).

There were 25 691 (12%) women with a record of antidepressant treatment. Women were predominantly prescribed SSRI (23 557, 92%) with TCA (1857, 7%) and other (2290, 8%) prescriptions being much less common. Of the women who had an SSRI prescription, there were 31% who had a record of depression (figure 1B), 31% who had a record of postnatal depression (figure 1B) and 33% who had depression symptoms (figure 1B). There were 6270 (27%) women with SSRI prescription *without* a record of either the depression diagnoses or symptom within a year after delivery. However, 4818 of these women had a record suggestive of depression or treatment *prior* to delivery leaving 1452 (6%) on SSRI treatment without a record suggestive of depression.

There were 6848 (3%) women with a record of referral for non-pharmacological treatment (figure 1A). Of the women receiving non-pharmacological treatment, there were 24% who had a record of depression (figure 1B), 22% who had a record of postnatal depression (figure 1B) and 29% who had depression symptoms (figure 1B), but 3064 (45%) with no records indicating depression, postnatal depression or depressive symptoms. However, 2041 of the the latter group of women had a record suggestive

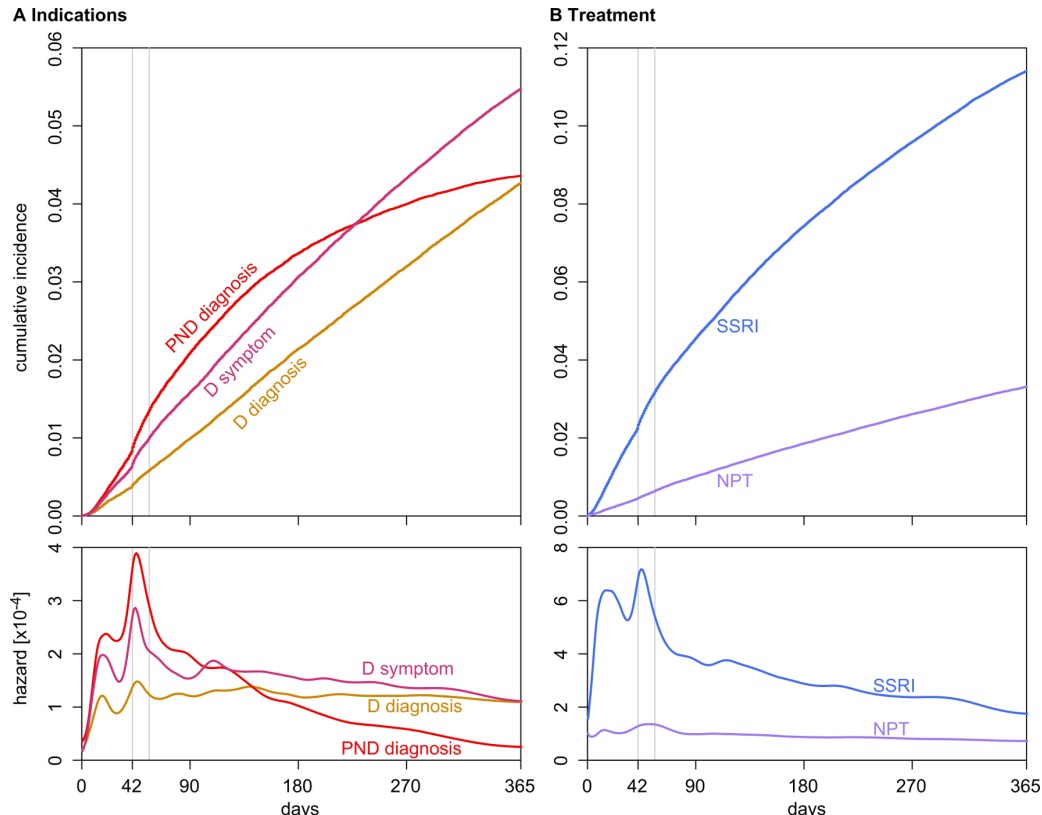

**Figure 2** Cumulative incidences and smoothed hazards for the records. Six and 8 weeks (6×7 and 8×7 days) are marked with a vertical grey line. Note the different y-axis scale for panels (A) and (B). D, depression; NPT, non-pharmacological treatment; PND, postnatal depression; SSRI, selective serotonin reuptake inhibitor prescription.

of depression or treatment *prior* to delivery leaving 1023 (15%) with a referral for non-pharmacological treatment, but without a record of depression. Of those with non-pharmacological treatment referral, 56% had SSRI prescription (figure 1B), whereas conversely only 16% with an SSRI prescription had a record of non-pharmacological treatment referral (figure 1B).

After the initial peak, the hazard for recording of postnatal depression and SSRI prescription show a markedly decreasing trend, while the other records show a relatively stable rate or slower decline (figure 2).

There were 64 283 (31%) women who had at least one record suggestive of depression or treatment at any time prior to delivery. The prevalence of depression and SSRI treatment *after* delivery was high among these women. Thus, there were 9666 (15%) with a record of depression or postnatal depression and 15 348 (24%) received SSRI treatment in the year after delivery. The figures were similar for women who have received SSRI treatment (n=40 178, 19%) at any time prior to delivery. Thus, there were 6940 (17%) with a record of depression or postnatal depression and 11 595 (29%) received SSRI treatment in the year after delivery.

### Age, social deprivation and time
Younger women were much more likely to have a record of depressive diagnoses or symptoms compared with women aged 30 years or older. For example, women

aged 15–19 years were nearly twice as likely to have a record of postnatal depression (RR, adjusted for social deprivation: 1.92 (1.76 to 2.10)) compared with women aged 30–34 years (table 1). There were no marked differences for women above the age of 30 (table 1). The pattern of SSRI treatment followed the same trends with nearly 1 in 5 women aged 15–19 receiving SSRI treatment in the first year after delivery (table 2) while for those aged above 30, it was 1 in 10 (table 2). Younger women were also more likely to receive non-pharmacological treatment than women aged 30 years or above (table 2).

The time to the initiation of SSRI treatment after the delivery has become earlier in the more recent years (figure 3). Thus, the initiation rate of SSRI treatment per 100 pregnancies (95% CI) at 8 weeks were 2.6 (2.5 to 2.8) in 2000–2004, increasing to 3.0 (2.9 to 3.1) in 2005–2009 and 3.8 (3.6 to 3.9) in 2010–2013. The overall rate of initiation of SSRI within the year after delivery, however, has not changed noticeably (table 2). The rates of non-pharmacological treatment have increased from 2.4 (2.2 to 2.5) per 100 pregnancies in 2000–2004 to 3.8 (3.6 to 3.9) in 2010–2013 (table 2). The recording of both depression diagnosis and postnatal diagnosis has decreased substantially over time while the recording of symptoms increased in the earlier time period, but have remained relatively constant since 2005 (table 1).

**Table 1** Rates and relative risk estimates of depression diagnosis, postnatal depression diagnosis and depression symptoms in the first year after delivery for 206 517 women who gave birth between 2000 and 2013

| | Depression diagnosis | | Postnatal depression diagnosis | | Depression symptom | |
|---|---|---|---|---|---|---|
| | Rate per 100 | Adjusted RR | Rate per 100 | Adjusted RR | Rate per 100 | Adjusted RR |
| Age group | | | | | | |
| 15–19 | 6.6 (6.1 to 7.1) | 1.64 (1.50 to 1.81) | 7.6 (7.1 to 8.1) | 1.92 (1.76 to 2.10) | 10.6 (10.0 to 11.2) | 2.10 (1.95 to 2.27) |
| 20–24 | 6.1 (5.9 to 6.4) | 1.59 (1.47 to 1.71) | 5.8 (5.5 to 6.0) | 1.49 (1.39 to 1.59) | 8.1 (7.7 to 8.4) | 1.63 (1.54 to 1.73) |
| 25–29 | 4.5 (4.3 to 4.7) | 1.22 (1.15 to 1.30) | 4.6 (4.4 to 4.8) | 1.21 (1.14 to 1.29) | 5.5 (5.3 to 5.7) | 1.18 (1.12 to 1.24) |
| 30–34 | 3.6 (3.4 to 3.7) | 1 | 3.8 (3.6 to 3.9) | 1 | 4.4 (4.3 to 4.6) | 1 |
| 35–39 | 3.5 (3.3 to 3.7) | 1.00 (0.93 to 1.06) | 3.5 (3.3 to 3.6) | 0.92 (0.86 to 0.98) | 4.3 (4.1 to 4.4) | 0.97 (0.92 to 1.02) |
| 40–49 | 3.1 (2.8 to 3.5) | 0.92 (0.81 to 1.03) | 3.2 (2.8 to 3.5) | 0.86 (0.77 to 0.97) | 4.7 (4.3 to 5.1) | 1.06 (0.96 to 1.17) |
| Calendar period | | | | | | |
| 2000–2004 | 5.7 (5.5 to 5.9) | 1 | 5.8 (5.6 to 6.0) | 1 | 4.4 (4.2 to 4.6) | 1 |
| 2005–2009 | 4.2 (4.0 to 4.3) | 0.71 (0.66 to 0.77) | 4.4 (4.2 to 4.5) | 0.73 (0.69 to 0.78) | 6.0 (5.8 to 6.2) | 1.31 (1.21 to 1.42) |
| 2010–2013 | 3.5 (3.3 to 3.6) | 0.58 (0.53 to 0.63) | 3.4 (3.3 to 3.5) | 0.56 (0.52 to 0.60) | 5.6 (5.4 to 5.8) | 1.21 (1.11 to 1.32) |
| Townsend Deprivation Index quintile | | | | | | |
| 1 | 3.2 (3.0 to 3.3) | 1 | 3.7 (3.5 to 3.9) | 1 | 4.1 (3.9 to 4.2) | 1 |
| 2 | 3.6 (3.4 to 3.8) | 1.14 (1.05 to 1.22) | 4.1 (3.9 to 4.2) | 1.09 (1.01 to 1.17) | 4.5 (4.3 to 4.7) | 1.07 (1.00 to 1.15) |
| 3 | 4.1 (3.9 to 4.3) | 1.26 (1.16 to 1.36) | 4.3 (4.1 to 4.5) | 1.12 (1.04 to 1.21) | 5.3 (5.1 to 5.5) | 1.19 (1.10 to 1.28) |
| 4 | 5.1 (4.9 to 5.3) | 1.51 (1.38 to 1.64) | 4.8 (4.6 to 5.0) | 1.20 (1.11 to 1.30) | 6.6 (6.4 to 6.9) | 1.42 (1.31 to 1.53) |
| 5 | 6.0 (5.7 to 6.3) | 1.69 (1.53 to 1.87) | 5.3 (5.0 to 5.5) | 1.26 (1.13 to 1.39) | 7.7 (7.4 to 8.0) | 1.56 (1.42 to 1.72) |

Adjusted by age group, calendar period, and Townsend Deprivation Index.
RR, relative risk.

The risk of having a record of depression, postnatal depression and depressive symptoms increased with increasing social deprivation (table 1), and similar patterns were observed for both SSRI treatment and non-pharmacological treatment (table 2). Thus, nearly one in seven women from the most deprived areas received SSRI treatment within the first year after delivery in contrast to 1 in 11 women from the least deprived areas (table 2). Supportive analyses suggest that the effect of age is, in general, stronger among the women *without* records suggestive of depression or treatment prior to delivery than among women *with* prior records (online supplementary appendix 1 STable 1-4). However, the effect of social deprivation and calendar time was similar in women with and without prior records of depression or treatment (online supplementary appendix 1 STable 1-4).

The women with early records (before 42 days after delivery) of depression, postnatal depression and depressive symptoms were more likely to have a prior record of depression or treatment (adjusted OR estimates of 2.43 (2.02 to 2.94), 1.58 (1.41 to 1.77) and 1.55 (1.37 to 1.76), respectively) (online supplementary appendix 1 STable 5) and have delivered more recently (especially for postnatal depression and depressive symptoms; respective adjusted OR estimates of 1.06 (0.87 to 1.28), 1.24 (1.08 to 1.42) and 1.65 (1.38 to 1.97) for the three records for the 2010–2013 calendar period against the baseline 2000–2005 period). The results were similar for women who had early records of SSRI treatment and non-pharmacological treatment (adjusted OR estimates of 3.02 (2.78 to 3.29) and 1.91 (1.62 to 2.27) for the prior record, respectively, and of 1.59 (1.46 to 1.74) and 1.36 (1.11 to 1.68) for the recent time period) (online supplementary appendix 1 STable 6). No clear trends were observed in the effect of social deprivation or age group, except an indication of the youngest age group having a higher proportion of early recording for postnatal depression diagnosis (adjusted OR estimate of 1.43 (1.17 to 1.75)) (online supplementary appendix 1 STable 5).

## DISCUSSION

We found that 11% of women who had given live birth had a record suggestive of depression in their primary care electronic health records within the first year after delivery. There were some peaks in recording of depressive diagnoses and symptoms and initiation of SSRI treatment soon after delivery (6–8 weeks), coinciding with the time of postnatal maternal check-up consultations although they continued to be recorded throughout the first year after delivery. The time to the initiation of SSRI treatment after the delivery has become earlier in the more recent years although the overall rate of initiation of SSRI within the year after delivery has not changed. Women with records suggestive of depression or SSRI treatment *prior* to delivery were more likely to have a

**Table 2** Rates and relative risk estimates of SSRI prescription and non-pharmacological treatment in the first year after delivery for 206 517 women who gave birth between 2000 and 2013

| | SSRI prescription | | Non-pharmacological treatment | |
|---|---|---|---|---|
| | Rate per 100 | Adjusted RR | Rate per 100 | Adjusted RR |
| Age group | | | | |
| 15–19 | 18.8 (18.0 to 19.5) | 1.78 (1.68 to 1.88) | 4.9 (4.5 to 5.4) | 1.55 (1.41 to 1.72) |
| 20–24 | 15.9 (15.5 to 16.4) | 1.54 (1.47 to 1.61) | 4.4 (4.1 to 4.6) | 1.38 (1.28 to 1.49) |
| 25–29 | 11.7 (11.5 to 12.0) | 1.18 (1.14 to 1.23) | 3.3 (3.2 to 3.5) | 1.11 (1.04 to 1.18) |
| 30–34 | 9.6 (9.4 to 9.8) | 1 | 2.9 (2.8 to 3.0) | 1 |
| 35–39 | 9.3 (9.1 to 9.6) | 0.99 (0.95 to 1.03) | 2.9 (2.7 to 3.0) | 0.99 (0.92 to 1.07) |
| 40–49 | 9.6 (9.0 to 10.1) | 1.01 (0.94 to 1.07) | 3.1 (2.7 to 3.4) | 1.05 (0.93 to 1.19) |
| Calendar period | | | | |
| 2000–2004 | 11.4 (11.1 to 11.7) | 1 | 2.4 (2.2 to 2.5) | 1 |
| 2005–2009 | 11.3 (11.1 to 11.6) | 0.97 (0.93 to 1.01) | 3.5 (3.3 to 3.6) | 1.43 (1.30 to 1.57) |
| 2010–2013 | 11.5 (11.2 to 11.7) | 0.96 (0.92 to 1.01) | 3.8 (3.6 to 3.9) | 1.54 (1.38 to 1.71) |
| Townsend Deprivation Index quantile | | | | |
| 1 | 8.9 (8.7 to 9.2) | 1 | 2.7 (2.5 to 2.8) | 1 |
| 2 | 10.0 (9.7 to 10.3) | 1.09 (1.04 to 1.14) | 3.0 (2.9 to 3.2) | 1.10 (1.00 to 1.20) |
| 3 | 11.3 (11.0 to 11.6) | 1.19 (1.14 to 1.25) | 3.3 (3.1 to 3.4) | 1.14 (1.04 to 1.25) |
| 4 | 13.1 (12.7 to 13.4) | 1.33 (1.25 to 1.40) | 3.8 (3.6 to 4.0) | 1.29 (1.17 to 1.42) |
| 5 | 15.2 (14.8 to 15.6) | 1.47 (1.38 to 1.57) | 4.2 (3.9 to 4.4) | 1.36 (1.22 to 1.52) |

Adjusted by age group, calendar period and Townsend Deprivation Index.
RR, relative risk; SSRI, selective serotonin reuptake inhibitor.

subsequent record and/or treatment *after* delivery. Likewise, of women with records of depression and treatment after delivery those with an *early* record (before 42 days after delivery) were more likely to have prior records of

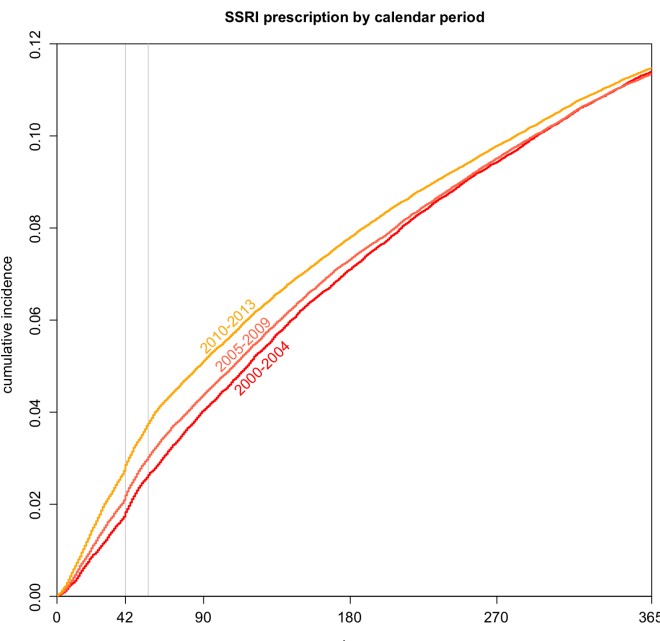

**SSRI prescription by calendar period**

**Figure 3** Cumulative incidence of selective serotonin reuptake inhibitor (SSRI) in three calendar periods. Six and 8 weeks (6×7 and 8×7 days) are marked with a vertical grey line.

depression or treatments than women with *later* records (after 42 days of delivery).

Younger women were more likely to have a record suggestive of depression compared with women aged 30 years or older and the pattern of SSRI initiation followed the same trend with nearly one in five women aged between 15 and 19 years receiving SSRI treatment in the first year after delivery. The risk of depression increased with increasing social deprivation, and similar patterns were observed for both SSRI treatment and non-pharmacological treatment.

### Strengths and limitations

A major strength of this study is that we have access to a very large sample of primary care electronic health records of women who gave live birth. These records reflect clinical practice in UK primary care and were made prospectively and therefore are not subject to recall bias. We considered a broad definition of depression based on clinical evaluation in the year after delivery as there are no specific guidelines to how it should be recorded in this period in primary care. Thus, we included women who had a specific diagnosis of postnatal depression, as well as women with records of depression diagnosis and symptoms, which may overestimate the number of women with postnatal depression compared with estimates based on a diagnostic interview and specific diagnostic instruments.

We are also aware that the indications for SSRI prescribing are broader than depression, and some

women in our study may have received SSRI for treatment for other indications, for example, anxiety. Yet, there is often an overlap between depression and anxiety,[18] and we chose, therefore, to include initiation of all SSRI prescriptions in our study. Our estimates of referral for non-pharmacological treatment were relatively low. This may reflect a limited accessibility to non-pharmacological treatment, but it is also important to be aware that often in clinical practice the booking system for referrals is not directly linked to electronic health records, and general practice staff will need to enter these referrals separately in the patient records. Furthermore, it is increasingly possible for women to self-refer themselves to psychological therapies through the 'Improving Access to Psychological Therapies' scheme in the UK (https://www.england.nhs.uk/mentalhealth/adults/iapt/). Therefore, it is likely that our study underestimates the actual referral rates for non-pharmacological treatments.

### Comparisons to existing evidence

Our summary estimate of postnatal depression, depression and symptoms of depression in the year after delivery (11%) was within the lower end of the range of previous prevalence estimates (10%–19%).[2–4] Gavin et al estimated point prevalence of minor and major depression was highest in the third month after delivery at 12.9%, although the CIs were wide.[2] The results of our study suggest a peak in depression records and antidepressant treatment within 6–8 weeks after delivery, coinciding with the time of postnatal check-up consultations.

Our findings of increase in the use of symptoms codes as opposed to diagnostic codes for recording of depression reflect previous findings on recording of depression in primary care in general.[19] Rait et al suggest GPs' coding may be linked to the perceived severity of depression, with symptom codes being used for milder depression. Alternatively, this move towards recording of symptoms and less specific terms may be perceived as less stigmatising for individuals.[19]

Nearly one out five women in our study had a record suggestive of depression and/or SSRI treatment records prior to delivery. Of these women, 17% had additional records of depression and more than a quarter received SSRI treatment in the year after delivery. Prior depression has long been recognised as one of the strongest risk factors for depression in the year after delivery.[1–3 20] We also found that women who sought help early (before 42 days after delivery) were more likely to have had a prior record of depression or treatment. They might be better at recognising the symptoms earlier on than women without prior experience. Thus, a qualitative systematic review of help-seeking barriers by Dennis and Chung-Lee concluded that lack of knowledge about postpartum depression or the acceptance of myths was a significant help-seeking barrier and rendered mothers unable to recognise the symptoms of depression.[21]

Many women discontinue antidepressant treatment in pregnancy.[22 23] A few studies suggest that these women are at higher risk of relapse,[24] but it is difficult to judge in observational settings and further research is needed to understand the role of antidepressant treatment in prevention of depression in the year after delivery.

Increased risk of postnatal depression among teenage mothers is well recognised with prevalence estimates as high as 26%.[25] Our study demonstrated that the level of recording of depressive diagnoses and symptoms continued to be higher for women right up to the age of 30, whereas no marked difference was found for women above the age of 30. Previous meta-analyses of postnatal depression have failed to recognise this 'L-shaped' difference in risk of postnatal depression with age.[1 3] In contrast to our findings, a recent Canadian study on women aged 20–44 years based on the Canadian Community Health Survey suggests that there is a 'U-shaped' relationship with age and postnatal depression. Thus, they found that the prevalence of depression in women who had recently delivered was significantly higher in women aged 40–44 years than in women aged 30–35 years (adjusted OR 3.72; 95% CI 2.15 to 6.41).[26]

There is some evidence that socioeconomic status is associated with prevalence of postnatal depression.[2 3 8 27] The results of a meta-regression analysis suggest that the prevalence of major depression is similar among socioeconomic status groups, but that minor depression may be more prevalent among lower socioeconomic status groups.[2] While we were unable to distinguish directly between diagnosis of major and minor depression, we observed a clear gradient with increasing level of deprivation across all measures of depression and treatments. An even stronger socioeconomic gradient in SSRI treatment was found among general population of adult women in the UK. Hence, women from the most deprived areas were 64% more likely to have been initiated on SSRI treatment compared with women from the least deprived areas.[28]

Our study reflects women's primary care electronic health records. For women to have records of depression, it requires that they have consulted their GP. However, some women may be reluctant to seek help and unwilling to disclose or discuss their problem because of fear of stigma, negative perceptions of them as a mother or fear that their baby might be taken into care.[6 21 29] Investigators and clinicians should also be aware of the potential differences in the way women express postpartum depression and that it may differ for women of different educational backgrounds.[30] Likewise, some healthcare professionals may miss or misdiagnose postnatal depression in the period soon after birth[6] and estimates based on primary care health records may underestimate the 'true' prevalence of postnatal depression. Our study clearly shows that for many women depression and depressive symptoms were 'picked up' and treatment initiated at the time of the maternal check-up consultation in accordance to guidelines on antenatal and postnatal mental healthcare.[29] Yet, our results have also revealed that depression is not limited to the immediate period after delivery and

emphasise the need for healthcare professionals to be alert to signs and symptoms of depression throughout the first year after delivery. Indeed, a recent systematic review suggested that screening postpartum women for depression may reduce depressive symptoms in women with depression and reduce the prevalence of depression.[31]

## CONCLUSIONS

More than 1 in 10 women had electronic health records indicating depression or depressive symptoms within a year after delivery, and more than one in eight women received antidepressant treatment in this period. Women aged below 30 and from the most deprived areas were at the highest risk of depression and most likely to receive antidepressant treatment.

**Contributors** IP, TP, SH, KRW and SK conceived the study. TP conducted the statistical analyses together with IP. IP and TP drafted the manuscript. All authors contributed to preparing the manuscript and have agreed to submit the final version of the manuscript. IP is the guarantor.

**Funding** SH, IP and KRW received funding from National Institute for Health Research (NIHR) School of Primary Care Research (grant 325). TP was funded by Academy of Finland (Finnish Centre of Excellence in Computational Inference Research Grant number 284642).

**Disclaimer** The views expressed in this publication are those of the author(s) and not necessarily those of the NHS, the National Institute for Health Research or the Department of Health.

**Competing interests** None declared.

**Patient consent** Not required.

**Ethics approval** The scheme for The Health Improvement Network to obtain and provide anonymous patient data to researchers was approved by the National Health Service South-East Multicenter Research Ethics Committee in 2002 and scientific approval for this study was obtained from IMS Scientific Review Committee.

**Provenance and peer review** Not commissioned; externally peer reviewed.

**Data sharing statement** As the data for this study have been bought under a licence, no data are available for sharing.

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
