## [Reviewer comments · BMJ Open]

ARTICLE DETAILS

TITLE (PROVISIONAL)	Depression, depressive symptoms and treatments in women who have recently given birth: UK cohort study
AUTHORS	Petersen, Irene; Peltola, Tomi; Kaski, Samuel; Walters, Kate; Hardoon, Sarah

VERSION 1 – REVIEW

REVIEWER	Kenji J. Tsuchiya Hamamatsu University School of Medicine, Japan
REVIEW RETURNED	28-Feb-2018

GENERAL COMMENTS	Comments: The large number of the study participants, who seem to be representative of the whole UK population, is an outstanding strength of this study. The estimates are thus reliable, and may be generalised. Clinical significance of this study is also of value. My comments and questions are as follows: 1. I am not quite clear about how the authors (or the administrators of the READ database) defined deression/postpartum depression and symptoms of depression. Are the prevalences estimated for each of these categories comparable to the reported prevalences reported in the literature? Are the assessments provided by the GPs to those with "symptoms of depression" reliable? What signs and/or symptoms are included in the "symptoms of depression"?2. How should we understand the difference between any sort of depression and use of SSRI? Although the authors stated the use of SSRI to be intended for those with anxiety disorder (pp. 9), can it be a sole account for the 45% (100% minus 55% SSRI prescribed with any form of depression: Fig 1b)?3. How did the authors count the number of those with depression before approximately 42 days after childbirth? What sort of women sought for help before 42 days as opposed to general referral patterns described in pp. 5? This is of interest as those with an early onset and with a later onset may reveal different background.4. The literature cited here appear to be outdated and/or insufficient in some parts. For instance, pp. 10, LL. -1 the metaanalyses were published 17 years ago; there are a number of studies that follow, which have reported the "L-shape". What are the bases for taking "lower doses may be prescribed for other reasons such as chronic pain"? (Also, what are the cutoff points for "lower doses"?)5. I appreciate it if the authors could provide some more details about the Townsend scores. Is the measurement based on income,
--

	education, occupation, residence, or anything else?
REVIEWER	MJ Saurel-Cubizolles INSERM U1153-EPOPé, Paris, France
REVIEW RETURNED	15-Mar-2018
GENERAL COMMENTS	The part of figure (1B) on the combination of diagnoses is not understandable. I am not sure the figures are useful. I don't understand why the adjusted RR related to Townsend deprivation index, estimated on all data (last rows table 1 or table 2) are higher than those estimated among women who had no prior record suggestive of depression and those estimated among women who had prior record suggestive of depression. For instance, for Depression diagnoses Townsend deprivation index quintile = 5 RR adjusted= 1.69 (1.53-1.87) for all women RR adjusted= 1.42 (1.24-1.64) for women who had no prior record suggestive of depression RR adjusted= 1.33 (1.20-1.47) for women who had prior record suggestive of depression. I was expecting the estimate on total data to be a weighted average of the estimates in the subgroups. Thank you for explaining this result. The most interesting result is the difference observed according to the deprivation indicator. The authors should highlight it and discuss it in more detail.

VERSION 1 – AUTHOR RESPONSE

R1:

The large number of the study participants, who seem to be representative of the whole UK population, is an outstanding strength of this study. The estimates are thus reliable, and may be generalised. Clinical significance of this study is also of value.

We thank you for your positive comments.

1. I am not quite clear about how the authors (or the administrators of the READ database) defined depression/postpartum depression and symptoms of depression. Are the prevalences estimated for each of these categories comparable to the reported prevalences reported in the literature? Are the assessments provided by the GPs to those with "symptoms of depression" reliable?

Depression and postpartum depression is defined in different ways in UK primary care. Some general practitioners (primary physicians) may prefer to use a Read code for depression whereas others may prefer to use a more specific Read code for postpartum depression and yet others may prefer to use symptom codes. It is therefore difficult to make a direct comparison to estimates to those reported in the literature. However, our overall estimates are comparable to those previously reported in the literature.

What signs and/or symptoms are included in the "symptoms of depression"?

Below we provide examples of the most commonly used codes for symptoms of depression.

1BT..11 *Low mood*
1B17.00 *Depressed*
1B17.11 *C/O - feeling depressed*
1BT..00 *Depressed mood*
1B1U.00 *Symptoms of depression*
1B1U.11 *Depressive symptoms*

2. How should we understand the difference between any sort of depression and use of SSRI? Although the authors stated the use of SSRI to be intended for those with anxiety disorder (pp. 9), can it be a sole account for the 45% (100% minus 55% SSRI prescribed with any form of depression: Fig 1b)?

Thanks for raising this issue. As we state in the discussion we are aware that the indication for SSRI prescribing is broader than depression and some women in our study may have received SSRI treatment for other indications for example anxiety. Yet, there is often an overlap between depression and anxiety and we chose therefore to include initiation of all SSRI prescriptions in our study.

We also believe that some GPs may omit to record another diagnosis of depression after delivery if the women already had such record. We have now estimated how often this occurred among women who were prescribed SSRI after delivery, but without records suggestive of depression.

Thus, we found of the 6,270 women with a prescription of SSRI without a record suggestive of depression within a year after delivery. Of these women, 4,818 (77%) had a record suggestive of depression or treatment prior to delivery. We have now included this information in the manuscript.

3. How did the authors count the number of those with depression before approximately 42 days after childbirth? What sort of women sought for help before 42 days as opposed to general referral patterns described in pp. 5? This is of interest as those with an early onset and with a later onset may reveal different background.

Thanks for raising this question, we have now examined the relationship between early (before 42 days) and late recording of depression and treatment according to age, social deprivation, calendar

year and prior records of depression. The main findings are that a prior record suggestive of depression or treatment and more recent calendar period is associated with earlier recording (before 42 days). We include a summary of these analyses in the result section and provide the results of the full analyses in appendix 2.

4. The literature cited here appear to be outdated and/or insufficient in some parts. For instance, pp. 10, LL. -1 the metaanalyses were published 17 years ago; there are a number of studies that follow, which have reported the "L-shape".

We have now updated the literature and for example we now include a reference to paper (Muraca GM, Joseph KS. The association between maternal age and depression. J Obstet Gynaecol Can JOGC J Obstet Gynecol Can JOGC. 2014 Sep;36(9):803–10.) that suggest the associations between age and postnatal depression is somewhat U shaped. However, a lot of the relevant literature is relatively old and there are a limited number of newer studies that deals with the questions.

What are the bases for taking "lower doses may be prescribed for other reasons such as chronic pain"? (Also, what are the cutoff points for "lower doses"?)

In the UK, treatment such as amitriptyline hydrochloride is also indicated for neuropathic pain at a lower dose (10 mg per day) and it is also used (although not licensed) for migraine prophylaxis (10 mg per day). In contrast the recommended initial dose for treatment of depression is 75 mg per day. (for further details please see <https://www.medicinescomplete.com>). Therefore, we only considered doses of 75 mg and above as treatment for depression.

5. I appreciate it if the authors could provide some more details about the Townsend scores. Is the measurement based on income, education, occupation, residence, or anything else?

We apologies for the omission of the details on the Townsend scores. We have now included following text in the paper "The Townsend scores is based on census data (2011) for car ownership, owner-occupation, overcrowding and unemployment in a patient's postcode."

R2:

Please leave your comments for the authors below

The part of figure (1B) on the combination of diagnoses is not understandable.

We apologies that we have not been able to describe the figure in sufficient detail. We now provide an example to help the reader to understand how the figure should be read. "For example, the figure illustrates that 82% of those who had a diagnosis of depression also had a prescription of a SSRI. On the other hand, 31% of those who had a prescription of SSRI had a diagnosis of depression."

I am not sure the figures are useful.

We feel the figure provides a lot of information about the interrelationship between the recording of depression and treatment and would like to keep the figure in the paper.

I don't understand why the adjusted RR related to Townsend deprivation index, estimated on all data (last rows table 1 or table 2) are higher than those estimated among women who had no prior record suggestive of depression and those estimated among women who had prior record suggestive of depression.

For instance, for Depression diagnoses

Townsend deprivation index quintile = 5

RR adjusted= 1.69 (1.53-1.87) for all women

RR adjusted= 1.42 (1.24-1.64) for women who had no prior record suggestive of depression

RR adjusted= 1.33 (1.20-1.47) for women who had prior record suggestive of depression.

I was expecting the estimate on total data to be a weighted average of the estimates in the subgroups. Thank you for explaining this result.

Below we seek to describe why the results differ from what one might have expected. The reviewer is right that the total data would have been close to a weighted average of the estimates in the subgroups if the distribution of individuals within each set quintile of Townsend scores had been the same in the two sub-groups. However, women with a prior record of depression are more likely to be more deprived than women without a prior record and the total data is no longer a weighted average (this is similar to the Simpson paradox except here the directions of the associations have not changed as might happen in extreme cases). We illustrate this with a fictive example below where women in group B is more deprived than women in group A.

Group	Townsend quintile	N	Number of events	Rate per 100	RR
A	1	15000	225	1.5	1.00
A	5	5000	175	3.5	2.33
B	1	500	40	8	1.00
B	5	1500	150	10	1.25
A+B	1	15500	265	1.7	1.00
A+B	5	6500	325	5.0	2.92

The most interesting result is the difference observed according to the deprivation indicator. The authors should highlight it and discuss it in more detail.

We do already touch on the difference observed according to deprivation in the discussion. Thus, we write:

“There is some evidence that socioeconomic status is associated with prevalence of postnatal depression. (2,3,8,27) The results of a meta-regression analysis suggest that the prevalence of major depression is similar among socioeconomic status groups, but that minor depression may be more prevalent among lower socioeconomic status groups. (2) While we were unable to distinguish directly between diagnosis of major and minor depression we observed a clear gradient with increasing level of deprivation across all measures of depression and treatments. An even stronger socio-economic gradient in SSRI treatment was found among general population of adult women in UK. Hence, women from the most deprived areas were 64% more likely to have been initiated on SSRI treatment compared to women from the least deprived areas.”

We will be happy to elaborate this if the editor feels it will strengthening the paper.

VERSION 2 – REVIEW

REVIEWER	Kenji J. Tsuchiya Hamamatsu University School of Medicine
REVIEW RETURNED	13-Jun-2018

GENERAL COMMENTS	Thank you very much for taking care of the comments. There remains a minor issue. In the abstract, it goes: Outcomes: Prevalence of postnatal depression, depression, depressive symptoms, However, under the header of "Data analysis" (pp. 6) it goes: First, we estimated the prevalence of any records directly suggestive of depression (postnatal depression, depression diagnoses, depressive symptoms)... . I am confused. Should the word "depression" in the "outcome" line of the abstract removed, as the term "depression" is considered to be collective and to lie at a higher order than "postnatal depression, depression diagnoses, depressive symptoms" etc? One more recommendation: I wonder if the fact that the data comes from "primary care electronic health records" (pp. 13) would better be clearly presented in the abstract, since the prevalence reported here cannot be compared to the ones in the literature because of this limitation.
--

VERSION 2 – AUTHOR RESPONSE

Please find our response to the points that the reviewer.

We have amended the abstract to match the description in the data analysis section. It now reads:

Prevalence of postnatal depression, depression diagnoses, depressive symptoms....

Further we have now clarified in the abstract that the data comes from primary care electronic health records.

VERSION 3 – REVIEW

REVIEWER	Kenji J. Tsuchiya Hamamatsu University School of Medicine
REVIEW RETURNED	29-Aug-2018
GENERAL COMMENTS	Thank you very much for the revision.